# Letting the World See through Your Eyes: Using Photovoice to Explore the Role of Technology in Physical Activity for Adolescents Living with Type 1 Diabetes

**DOI:** 10.3390/ijerph19106315

**Published:** 2022-05-23

**Authors:** Diane Morrow, Alison Kirk, Fiona Muirhead, Marilyn Lennon

**Affiliations:** 1Department of Computer & Information Sciences, University of Strathclyde, Glasgow G1 1XQ, UK; marilyn.lennon@strath.ac.uk; 2Department of Psychological Sciences & Health, University of Strathclyde, Glasgow G1 1XQ, UK; alison.kirk@strath.ac.uk (A.K.); fiona.muirhead@strath.ac.uk (F.M.)

**Keywords:** type 1 diabetes (T1D), physical activity, adolescent and parent, technology, HCI diabetes

## Abstract

This paper qualitatively explores how technologies and physical activity are experienced by adolescents with type 1 diabetes. Type 1 diabetes is a life-threatening autoimmune condition, which is highly prevalent in young children. Physical activity is underutilised as part of treatment goals due to multifactorial challenges and lack of education in both the family setting and across society as a whole. Using photovoice methodology, 29 participants (parents and adolescents), individually or as dyads, shared and described in reflective journal format examples of technology and physical activity in their lives. In total, 120 personal photographs with accompanying narratives were provided. The data were thematically coded by the researcher and then collaboratively with participants. Four key themes (and 12 subthemes) were generated including: (i) benefits of technology; (ii) complexity and difficulty; (iii) emotional impact; (iv) reliance and risk. Findings demonstrate that current technology does not address the complex needs of adolescents with type 1 diabetes to enable participation in physical activity without life risk. We conclude from our findings that future technologies for supporting engagement in physical activity as part of diabetes management need to be: more interoperable, personalised and integrated better with ongoing education and support.

## 1. Introduction

Type 1 diabetes (T1D) is a serious life-long, life-threatening autoimmune condition which is commonly diagnosed in youth. World prevalence of T1D continues to rise and there is currently no cure. According to the International Diabetes Federation, over 1 million children and adolescents under the age of 20 live with T1D [1]. It is a chronic condition that requires acute attention including frequent insulin medication and continuous blood glucose monitoring. Short-term acute risk can be severe and life-threatening. Longer-term complications include macro- and microvascular deterioration of cardiovascular systems and organ health. Treatment of the condition involves careful and continual self-management of blood glucose to reduce short-term acute risk and avoid long-term development of these complications. Healthy living including regular physical activity (PA) is known to help improve health and wellbeing, and reduce long-term complications linked to T1D [2,3,4].

Digital technologies for both type 1 diabetes (T1D) self-management and physical activity (PA) are developing at pace. In both fields, for example, technology is moving towards using sensors, artificial intelligence and algorithms to predict, monitor and track (and in some cases regulate) both movement and glucose variation. Insulin pumps connected to glucose sensing devices with control algorithms, often referred to as closed-loop systems, are being testing for effectiveness at the University of Cambridge, UK. They have produced the first commercially available closed-loop system in the UK. Testing and reporting of the closed-loop system have been ongoing for over ten years, showing significant benefits for health, wellbeing and glycaemic outcomes in young and adult populations [5,6,7,8,9,10]. Hands-free interaction with technologies such as this may assist in reducing the user burden, often found whilst being active [11]. Increased adoption of physical activity technologies which can detect physical activity is noteworthy but still requires further research [12]. After the pandemic, it is expected that more devices and also virtual fitness will rise in popularity [13]. However, little is known about the nature of the use of PA technology to support physical activity (PA) of adolescents living with type 1 diabetes, and the role of parents/caregivers as co-users. Ramifications of these technologies in the context of PA are yet to be fully explored. 

Research reports that children and adolescents living with T1D are not meeting the advised guidelines of 60 min of moderate to vigorous physical activity (MVPA) per day [14,15,16]. Physical activity behaviours can form during childhood and so identifying and overcoming potential barriers to PA participation in early years is important [17]. Physiological studies link a lack of PA to poor health outcomes in the general population and for people with T1D [18,19,20,21]. However, introducing PA brings additional challenges to the self-management of T1D. Understanding energy expenditure and careful insulin adjustment/titration for PA requires ongoing, appropriate, individualised education and behavioural support, most of which is not well understood and requires intensive self-management [22,23]. To show just how recent our understanding is of PA and T1D, one latest study explored the effects of prolonged aerobic exercise (60 min moderate) on blood glucose of youths with T1D. This was led by a team of experts in Canada [24]. This team reported that, out of 120 youths tested, only 4 had stable blood glucose during an exercise period, with over 93 having a significantly clinical drop in blood glucose. Youths measured had a wide variation in their pre-exercise glycaemia. However, the suggested conclusion from the study showed that having higher pre-exercise blood glucose was only “marginally protective” against exercise-induced hypoglycaemia. It is widely known that hypoglycaemia, and the fear of, exists as a major barrier for this age group. Finding the best approach to support PA is a challenge in the care of adolescents with T1D. MacMillan, et al. suggest how to implement improved communication, education and clarity of teachers’ roles and responsibilities in schools. This research shows that teachers also have fears related to T1D, and this too can create a barrier to PA support [25].

To design successful and meaningful digital interventions or tools that promote and/or support adolescents’ engagement in PA, more needs to be understood about the daily experiences of adolescents living with technology and T1D. This paper presents the findings from a 6-month photovoice study with 29 participants (adolescents living with T1D and their parents). The results are discussed in terms of (i) the captured lived experiences of adolescents living with T1D (and their parents) in relation to their engagement with PA and the role that technology plays in this and (ii) the use of photovoice for empowering adolescents living with life-long health conditions by providing space for their voice to be heard within research and digital intervention design.

## 2. Materials and Methods

This study design involved the use of photovoice over 6 months to understand experiences of technology and PA while living with T1D. Participants (N = 29) were adolescents with T1D and their parents (see Table 1 for details). Participants were recruited as co-researchers from the outset. Figure 1 illustrates the study procedures. The study was conducted under Departmental Ethics Board approval (Ethics ID 1083).

### 2.1. Design

This was an exploratory qualitative study of lived experience. Participants were invited via social media adverts to upload photos (over 14 days via their smartphones to private social media messages with the researcher) with accompanying reflective text to represent their day-to-day experiences specifically relating to the role of technology in engaging (or not) with PA.

### 2.2. Participants

Participants were invited to participate with the help of a parent, dependent on age (<16 years), or on their own or as parents/caregivers (see Table 1).

### 2.3. Procedures

Participants were invited via social media adverts to upload photos (over 14 days via their smartphones to private social media messages with the researcher) with accompanying reflective text to represent their day-to-day experiences specifically relating to the role of technology in engaging (or not) with PA. Data (photos and text) were immediately transferred to a secure university password-protected server location. One-to-one video discussions (collaborative coding sessions) were later also recorded with consent (audio and visual), for transcription as supplemental data, along with additional researcher field notes. COVID-19 restrictions on social meetings were in place in the latter stage of this study. However, remote video conferencing was found to be a required but acceptable study adaptation. 

Figure 1 shows the outline of the study design and procedures involved.

### 2.4. Ethics, Recruitment and Training

Ethical approval was granted for this study (1083).

Participants were recruited via adverts posted on social media (Twitter and Facebook) from February to July 2020. On Twitter, hashtags such as #T1d #gbdoc (Great British Diabetes Online Community), #type1diabetes and #doc (Diabetes Online Community) were used to target recruitment. Facebook groups relating to parents and caregivers of children living with T1D were also approached. To comply with ethical regulations, if the adolescent was under 16, parents were asked to submit the data on their child’s behalf, and include consent and assent signatures by wet ink (photograph emailed), or email consent. If the adolescent was 16–18 years old, their lone submission was agreed through consent, using remote methods as described above.

Data generation and communication were staggered throughout 6 months, depending on the date of received consent. Participants were told that data would be removed from private messenger or email once analysis was performed. The following four research areas were constructed as guiding topics for participants. These were communicated to participants to support data collection.

What technology are you using?What physical activity are you doing?What support is involved for you to be physically active?What other related impact does physical activity have on your lives?

Support and training were provided throughout the study. An orientation video was designed to provide the research aims, photovoice definition, brief photography skills (zooming and capturing) and the use of language as narrative (https://youtu.be/bhvFEBDuAxk, accessed on 1 February 2020). The use of the ‘SHOWeD’ terms helped co-researchers to facilitate self-motivation in capturing their lived experiences [26]. This mnemonic allowed for co-researchers to think about their image in a critical sense, raising questions such as: What do you see here? What is really happening here? How does this relate to our lives? Why does this situation exist? What can we do about it?. The environment in which photos were taken was chosen by the co-researchers. The image was chosen by them and the accompanying words were written solely by them. A private Facebook group for co-researchers—providing helpful tips, and neutral examples of photographs with narrative—was created for those who wished to join. Of consented participants, 60% joined the group and 7 individuals commented on posts. Through this training, a coaching relationship between the researcher and co-researchers developed.

### 2.5. Data Analysis and Shared Meaning

Photographs and “voice” (stories and narratives accompanying the photos) were analysed following the six phases of reflexive thematic analysis created by Braun and Clarke [27]. Initial coding was inductive, data-driven and involved assigning codes in preparation for later discussion with co-researchers for approval/change. Data analysis focused on co-researchers’ experiences of interactions with technologies, the role of technologies and their PA behaviours, including what type of activities they are involved with and the importance of the technology and other people in enabling and empowering their lifestyle.

Analysis was performed both collaboratively and solely by the researcher, depending on co-researchers’ choice. During collaborative analysis, whilst looking at each photograph and narrative, co-researchers were asked to think about why they shared a particular photograph. Promoting and developing critical thought was encouraged using the ‘PHOTO’ mnemonic shown below to label meaning and define interactions and experiences during this collaborative coding process.

P—Describe your Photo

H—What is Happening in your photo?

O—Why take a picture of this?

T—What does this photo Tell us about your life?

O—How can this photo provide Opportunities to help your life?

Co-researchers reviewed their data with the researcher during video interviews. Theme refinement and language changes occurred during this analysis stage (see Table 2 for an example). 

Moving from individual photographs and narratives across their overall data helped the co-researchers identify possible themes (theme development is shown in the Appendix A). This stage was considered important in terms of co-design and co-participatory elements of the study. For participants who agreed to online video interviews (see Figure 1), their input, reflections and opinions on the choice of initial codes were critical for establishing trustworthiness in the findings. This was taken into consideration when reporting the final themes. For co-researchers who did not agree to video discussion, an email was distributed with their coded data for member/sense checking. If co-researchers did not reply to the email, the codes were assumed acceptable and further rounds of analysis were conducted with research team members. After the second cycle of collaborative discussions, themes generated were noted and further analysed by the research team independently before a further round of approval by co-researchers. Any discrepancies were reconciled with co-researchers and analysis was agreed upon. Table 2 provides an example of theme refinement through collaborative video discussion.

Viewing images repeatedly with associated narrative and noting potential codes and themes was a process which was also conducted in non-participatory analytical coding sessions with research team members. The synthesis of comparing and combining the participatory and non-participatory analysis using a blending of thematic analysis is often used in photovoice research [28]. Taking informal research notes and reflecting on the images allows for early descriptions and labels which form codes. See Table 2 for examples of this process. A continual grouping of codes into themes, until there were no more themes in the data, was performed. Defining higher-order themes was explored by clustering codes into contextualised groups. Using Nvivo software (www.qsrinternational.com, accessed on 1 July 2020) during the process assisted with the finalising of results.

## 3. Results

In total, eight dyads (parent and adolescent with T1D) took part as well as five parents individually and eight adolescents individually—totaling 29 co-researchers. One participant data set was represented by two sisters who were both living with T1D. Table 1 describes co-researchers who took part. The average age of the adolescents was 12.9 years. In total, 120 photographs were submitted with an accompanying narrative totaling 8934 words. Example data are provided to represent the 12 subthemes in Figure 2, Figure 3, Figure 4 and Figure 5.

### Photographs with Narrative

We asked co-researchers to share their experiences relating to the four main topics/questions described in Section 2.3 above. We present these descriptively here in terms of each of the four key themes that were generated from the data analysis phase. Descriptions include what each of these means for the future design of technology to support diabetes management in Section 4.


**Theme 1: Benefits of Technology (Figure 2)**


One of the key themes was people’s experiences of and expectations for the benefits of technology in diabetes management and engagement with PA. Figure 2 shows some examples of photos with narratives that illustrate three of the sub-themes within this theme.

(a) Subtheme “**communication and immediate support**” relates to co-researchers’ narrative surrounding being able to contact parents/adolescents using smartphones by text, or message to ask for advice. This subtheme also relates to the benefit of communication in supporting independence for PA—being able to engage in activities with a degree of freedom from parental involvement. However, a paradoxical relationship exists as described in Figure 4.

(b) Subtheme “**passivity**” was identified as a subtheme when co-researchers spoke of their technology adjusting insulin dosage frequently, without them having to interact with the technology. Included in this passive experience was a sense of trust that technology would suspend delivery of insulin if the user was moving towards hypoglycaemia. This “just in time” effect of hypoglycaemia prevention is further supported by the user feeling trust in the system to alert them. Allowing them to enjoy being active through a hands-free system which would “beep” or “alarm” if they were “hypo” or “hyper”. However, as Figure 5 shows, there also exists a paradoxical experience with these elements of current technology.

(c) Subtheme “**glanceability**” describes looking at graphs and users being able to resume being active after a glance. The images show different types of technology used when living with T1D.


**Theme 2: Complexity and Difficulty (Figure 3)**


The second key theme captured the essence of complexity and difficulty which related to co-researchers describing a variety of challenges including transitioning to independence, the uncertainty of how to manage T1D and PA, and parental burden experienced.

(a) Subtheme “**transitions from parent**”. Co-researchers often mentioned being accompanied by parents and collaborating using messaging, alerts and often parental involvement from the side. This data often represented complex decisions, and difficulties surrounding comprehension, social identity and freedom from parents.

(b) Subtheme “**uncertainty and PA-induced hypos**”. This relates to data in the photos and narrative in which co-researchers expressed difficulty and complexity surrounding the prediction, prevention and treatment of hypoglycaemia related to PA. The example provided gives deeper meaning to the psychosocial aspect of being included in sports, and the benefits of being active despite a “rollercoaster” of glycaemic variation to experience.

(c) Subtheme “**parental burden in facilitation**”. This sub-theme represents data that were provided by parents as co-researchers who often accompany their child to events; they interact with their child’s diabetes technology before, during and after activity. They engage with technology on behalf of their child. Parents encourage and support their child to be active—whilst engaging in prediction management behaviours “behind-the-scenes”. This was spoken of by many parents in the study as a difficult but necessary part of their child’s PA experiences.

All three subthemes in this section relate to complexity and difficulty across technological, family and the wider context in terms of highlighting deeper context and age-related consideration of behaviours.


**Theme 3: Emotional Impact (Figure 4)**


Key theme 3 relates to the data in which co-researchers describe their feelings and emotions. Data representing parental emotions and adolescent emotions were rich and powerful. Many emotions pointed towards a relationship between navigating transition, support from others and the role of technology for managing T1D in interrupting or impeding the ability to engage in PA with ease, spontaneity and fun. A few co-researchers spoke of their gratitude for technology. There were instances of dualistic emotions expressed in the same data item, representing mixed emotions. The language used by co-researchers represented worry, fear, anger and resilience.

(a) Subtheme “**confidence through support**” is illustrated and represents the perceived sense of empowerment experiences and the confidence building involved when PA experiences are supported by others. Co-researchers spoke of sports groups, friends, social media support groups and elite sports camp experiences as shown in the example below.

(b) Subtheme “**resilience and positivity**” captured the range of positive emotional experiences reported, for example, when co-researchers spoke of finding strength and a sense of determination.

(c) Subtheme “**teen—negative emotions**” captures the experiences relating to negative emotional experiences experienced specifically by teens. In the example given, the teen expresses hate after experiencing hyperglycaemia and requiring medical support, which encroached on his ability to take part in PA.


**Theme 4: Reliance and Risk (Figure 5)**


The fourth key theme captures the core concept spoken by co-researchers about their experiences with health risks, especially when technology required intervention, troubleshooting or replacement.

(a) Subtheme “**tech fails and severe life risk**”. This subtheme represents co-researchers relating to when technology fails. Particular areas of failures related to cannula failings, glucose sensors falling off, loss of signal from glucose sensors and smartphone app signal loss. Additionally, there were mentions of adolescents not hearing alarms. In this context, adolescents were being active and either did not hear the alarm, or they heard it but ignored it to continue their activity. There was also a mention of reliance on a parent or other to assist them.

(b) Subtheme “**trust and vigilance**” which provides interpretations of a sense of trusting technology and the relationship this creates with vigilance in the user. In the example shown, the parent felt that remote monitoring after an active day gave a clear indication of life risk, despite technologies’ best efforts to stave off exercise-induced hypoglycaemia.

(c) Subtheme “**alerts and vigilance**” points to the relationship between technology alerting the user to health risk and the complex relationship with the choice/ability to attend to these alerts through vigilance. In the example provided, the user felt their attention was not captured by the alerts enough to warrant behaviour change at that time. When they experienced further sequential alerts and a serious level of hypoglycaemia, the user then decided to take action to treat their hypoglycaemia.

## 4. Discussion

This study involved using photovoice to understand lived experience of adolescents living with T1D with a focus on the role of technology in engaging in PA. Our intention was to understand these interactions and the needs of the adolescents and their parents. The study methods aimed to help adolescents and their parents become active agents to identify their needs, challenges or successes based on real lived experiences through the use of photovoice. The four key themes generated were (i) benefits of technology; (ii) complexity and difficulty; (iii) emotional impact; (iv) reliance and risk. A discussion of the themes set in the wider context of published literature about both technology and PA and psychosocial factors is offered. We also share our recommendations for the future design of technology to support the lived experiences of adolescents engaging in PA while living with T1D.

### 4.1. Benefits of Technology

There is a range of technologies involved in diabetes management, including insulin pumps with associated cannulas, glucose meters and glucose sensors, from many different manufacturers and models. Passive technologies (using controller programming and artificial intelligence) were perceived to be of significant benefit. People in the study commented on how lucky they felt to be able to access some of the devices. Others talked about how hands-free dose adjustment could play a role in enabling adolescents to be free to take part in PA in that for people involved in competitive sports, being able to glance hands-free during play/activity was a significant enabler. This aligns with previous reports linked to a positive quality of life when the technology was tested for home use in early trials [29].

Being able to communicate and/or share blood glucose data with others via the devices is also perceived to be of huge benefit. Glancing at the info on a smartphone or other device provided peace of mind and a greater sense of independence both for the adolescent and/or for the parent. Peace of mind was also shown as a key theme by Burckhardt, et al. [30] in their study related to remote monitoring. This overlap of easing psychosocial burden is mirrored in our study.

A lack of technology specifically to support PA was reported in our study. Only two participants from a possible 29 spoke of engagement with wearable physical activity technology, one of which showed a visualization of blood glucose data transmitted from a glucose sensor device. Benefits of such technology were however depicted in terms of feeling “freedom” to go to work (the parent) and trusting in the data whilst sleeping at night, for example. Further research is needed into how physical activity technologies can support adolescents living with T1D and how to support engagement with these technologies.

### 4.2. Complexity and Difficulty 

There are many factors which relate to the challenges experienced in living an active lifestyle with T1D. Adolescents long to become independent from parents during such activities to be free to enjoy PA without being called from the sidelines to eat or drink. Parents however often feel like the gatekeepers, facilitating and/or being responsible for “safe play” or “safe participation” in sports given the potential risk to life. Parents also appear to be very mindful of the amount of forward planning that is required when supporting upcoming periods of PA or specific events. This gatekeeper effect was noted in previous research [31]. There is also evidence for lots of behind the scenes planning and calculations by parents. Uncertainty over how to prevent exercise-induced hypoglycaemia, or the correct and safest strategies to cope with dose adjustment to prevent late-onset hyperglycaemia, is prevalent. Overcompensating with food changes, dose adjustment guesstimates and managing the uncertainty of glucose variation occur despite their best efforts to create a safe PA experience for their child. There is a need for ongoing consideration of the family’s role in supporting active lifestyles of adolescents living with T1D.

### 4.3. Emotional Impact

Parents who use glucose-sensing devices linked to insulin pump delivery systems spoke of fear of PA-induced hypoglycaemia. Waiting by the swimming pool, staying at the cricket pitch and attending school activities despite using advanced sensing technologies. In critically analysing this relationship, a sense of vigilance was related to experiences of technological failures, and vigilance is therefore an important factor. Choudhary, et al. [32] stated that confidence in the use of sensor-augmented insulin pump systems does exist, and sensitivity of the signal to reduce insulin delivery can be a success, but our data show that the lived experiences of this can vary from device to device and person to person. Parents talked of feeling grateful for the technology but also spoke of technological immaturity. Loss of signal, loss of data and lack of connected devices were described as “scary” by parents, and “worrying” by teens. Our findings show there is a lack of trust in the technology, which further contributes to the psychological burden of worry, fear and anxiety. Troubleshooting these failures and navigating risk is an important part of daily diabetes management and technology does not yet seem to currently negate all the psychological and emotional burdens. As suggested by Borus and Laffel [33], unless closed-loop systems are offered and tested on a wider scale, these issues related to burden and sub-optimal glucose control will continue to be experienced.

Visual displays of, for example, glucose data on smartphone apps, wearable devices and insulin pumps were reported as a positive experience related to reliance. The benefit of a good display design for glancing at real-time data can provide an opportunity to respond to and treat a life-threatening event (hypoglycaemia or hyperglycaemia) in a timely fashion. Setting alarms on smartphones and using social media to connect with others were shown to be useful.

Nocturnal exercise-induced hypoglycaemia is a frequent fear and concern, and technology can provide a trustworthy alert function which allows parents to sleep better and have peace of mind. However, our data reveal that the technology may not be advanced enough yet to prevent nighttime risk and parents in our study reported frequent sleep disruption and emotional impact. This continues from existing work which looked at nocturnal hypoglycaemia associated with fear [34,35]. New research suggests a move toward assessing parental sleep disruption as a call for intervention support, showing a clear relationship between T1D management and sleep disruption for both the person living with T1D and the parent [36].

### 4.4. Reliance and Risk

In relation to diabetes technologies, experiences reported in this study often related to failed glucose sensors, cannulation or injection site failure and skin issues (allergy to adhesion and adhesion reliability). Insulin pump failures and encroachment on PA were also reported. Co-researchers in our study experienced managing device failures by troubleshooting during or after PA. There were stories shared that portrayed frustration at device failures and the subsequent failure to be active due to these events. Often the narrative shared expresses emotions in relation to reliance and risk, for example, the adolescent feeling angry. Implications for disruption to play, type and duration of activity were revealed in our study. There are some similarities in our study to previous research which addresses the type and duration of PA and the barriers associated with use of technologies [37]. Similarities included water exposure, coming into contact with others during activity and adhesion of the devices on the body.

Further interruption to PA through risk as a factor includes alarms, alerts acting as warnings and deterrents. Alerts and alarms are intended to offer peace of mind and communication to prevent severe risk. We see from our study, however, that this deterrent does not always have the desired effect, especially if adolescents are enjoying their activities, do not hear the alerts or alarms or do not want to be interrupted. The technology can in this sense be perceived to be an intrusion and can result in ignoring the device or alarms. Adolescents in our study used strong emotive words to describe these interruptions and spoke of feeling “hate”, “angry” and “annoyed” when interrupted by their insulin pump alarms or alerts. Not hearing alarms or avoiding alarms due to fear of exclusion was also reported across our data. This finding aligns with other reports of “alarm fatigue” by Shivers, et al. [38].

In terms of peace of mind versus fear of hypoglycaemia, the key theme of “reliance and risk” captures the co-researcher’s varied experiences. Experiences of reliance relate, on the one hand, to the critical, life-saving aspects of technology (important warnings of alarms) and, on the other hand, to a strong sense of fear when the technology requires troubleshooting or negatively disrupts daily life.

### 4.5. Strengths, Limitations and the Use of Photovoice

The visual data (photos) contributed to the research questions our study proposed by providing space to empower adolescents and their parents to tell their stories through their eyes. They often shared visual depictions of their technology, which may normally be hidden beneath clothing, or hidden due to feelings related to identity. Alleviating the mystery behind living with T1D. Their critical analysis of their own lived experience brought to the surface a vivid and lucid powerful examination of their lives. Coding data with co-researchers facilitated a collaborative framework and generated trustworthiness in the data and co-validation of meaning in the photos and narratives that represented lived experiences of our participants. Guided discussion diminished perceived control and power relations in the research and acted as a bridge between academics and the young people involved in the research study [39]. Limitations of using photovoice from our experience are related to prior knowledge, training and developmental ability of the adolescent to think critically, as discussed previously by Latz [40]. There is currently a lack of standardised quality measures in place to guide and replicate a photovoice study. Previous participatory research explored cognitive appropriateness, provision of choice and co-researcher identities within photovoice studies to increase feelings of power and reduce stigma for adolescents with T1D [41]. The first author researcher approached learning and training from a combination of reading materials, peer-reviewed journals, books and video footage to explore how best to lead civic engagement, engage youth and facilitate independent thought [42,43,44,45]. In summary, photovoice allows for collaborative practice, empowerment, ownership and identification of behaviours which is important to communicate to providers of healthcare and technology design.

## 5. Conclusions

Our findings demonstrate that current technology does not yet address the complex needs of adolescents living with T1D to enable participation in PA without health risks.

We offer the following recommendations based on our analysis for how technology can be improved to better support these adolescents and their parents/caregivers:(i)Diabetes-monitoring technologies and PA devices need to be more interoperable and work more seamlessly together (e.g., accelerometry data and glucose variability). Combining these technologies should seek to incorporate the benefits of “hands-free” experiences using glanceability and data sharing through trusted channels. Interoperability will reduce the amount of devices that people interact with in order to make informed choices about T1D management during periods of PA.(ii)Interventions need to facilitate and promote peer support which considers adolescents and parents/caregivers as central, active agents in a workshop-led or digitally based education. These support mechanisms are important for continuing to build agency and empowerment to understand technology, as it changes and develops rapidly. Parents also require ongoing support as caregivers and intense users of the technology.(iii)A focus is needed on psycho-educational support for navigating independence for both those living with T1D and their parents/caregivers, especially during transition periods. Physical activity is often performed away from home or in the community where teachers and coaches often lack knowledge of the effects of PA on T1D. This often caused frustration, and anxiety for both parents as caregivers and adolescents living with T1D.(iv)Digital educational content should be personalisable to person-specific technological choices, i.e., pump users, CGM users, pens/meters, and to teachers and healthcare professionals with accompanying tailored PA advice. Consumer power for choices of technology allows for different technologies to treat T1D. These can impact the choice, duration and type of PA taken.

## Figures and Tables

**Figure 1 ijerph-19-06315-f001:**
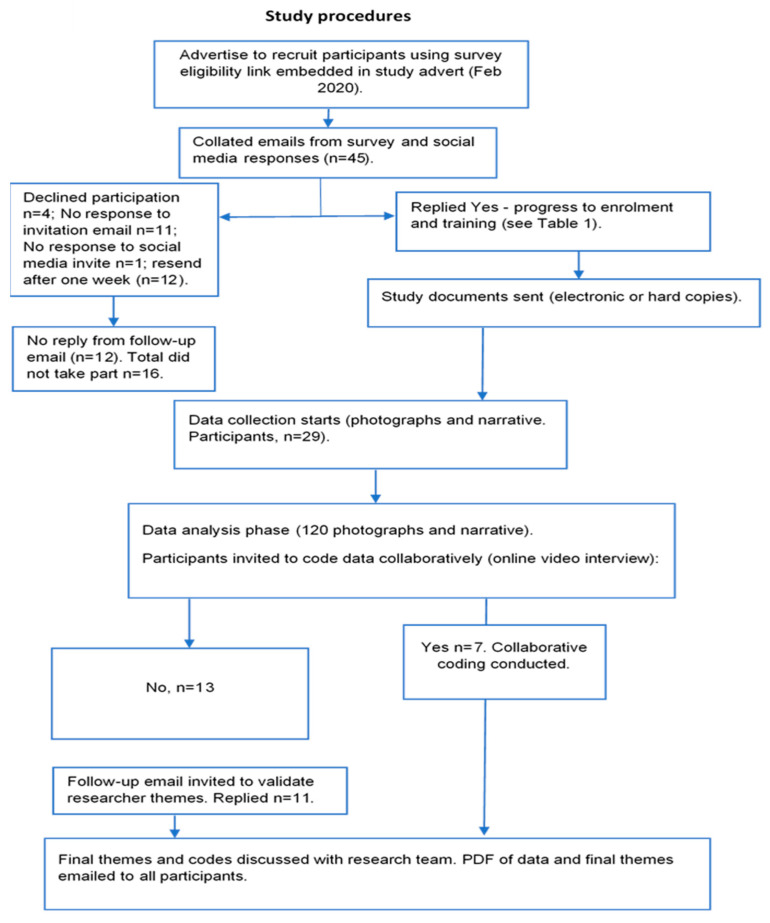
Flow chart showing study design; recruitment; collection and analysis phases.

**Figure 2 ijerph-19-06315-f002:**
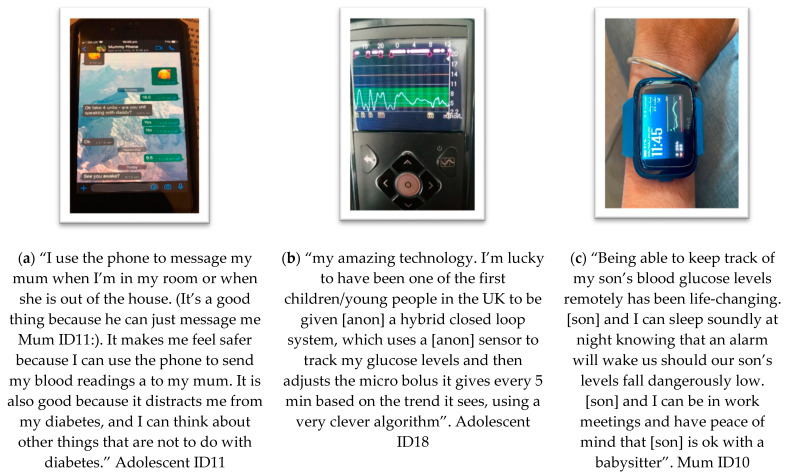
Photovoice submissions representing theme 1, benefits of technology, showing data interpreted as the following sub-themes: (**a**) communication and immediate support; (**b**) passivity; (**c**) glanceability.

**Figure 3 ijerph-19-06315-f003:**
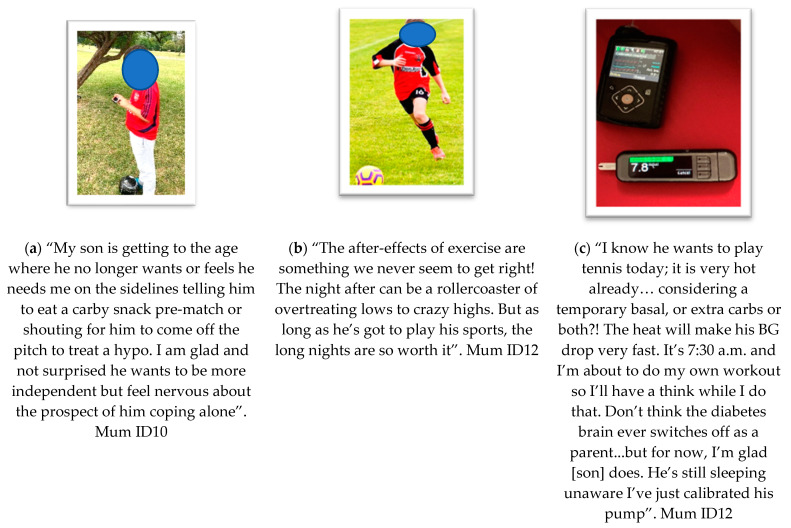
Photovoice submissions representing theme 2—*Complexity & difficulty*, showing data from interpreted as the following sub-themes: (**a**) transitions from parent; (**b**) uncertainty & PA-induced hypos; (**c**) parental burden in facilitation.

**Figure 4 ijerph-19-06315-f004:**
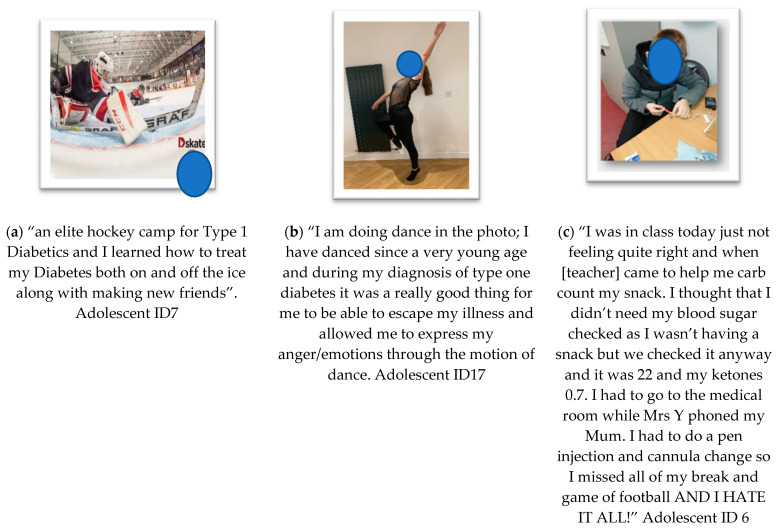
Photovoice submissions representing theme 3—*Emotional impact*, showing data interpreted as the following sub-themes: (**a**) Confidence through support; (**b**) Resilience & positivity; (**c**) Teen—negative emotions.

**Figure 5 ijerph-19-06315-f005:**
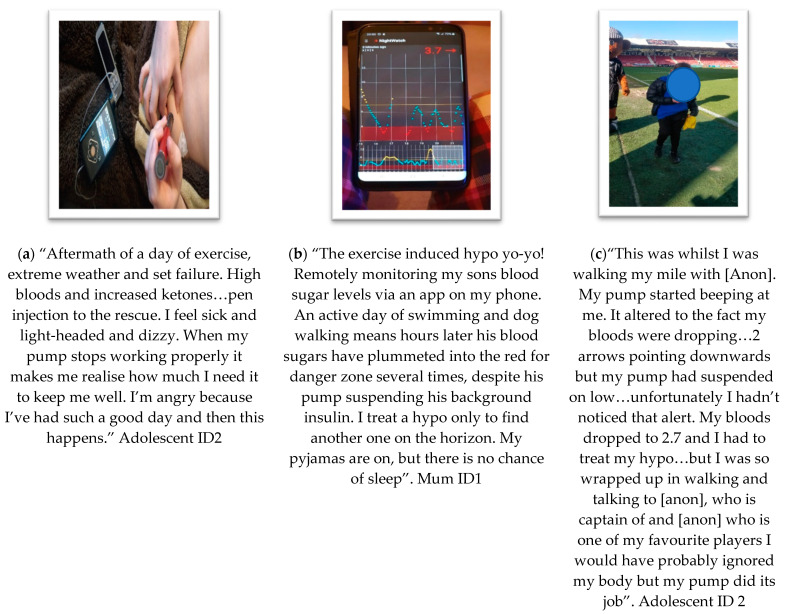
Photovoice submissions representing theme 4, reliance and risk, showing data interpreted as the following sub-themes: (**a**) tech fails and severe life risk; (**b**) trust and vigilance; (**c**) alerts and vigilance.

**Table 1 ijerph-19-06315-t001:** Participant descriptions showing variations of adolescent and parents acting in partnership to submit data, and adolescents submitting on their own, showing age and gender.

Participant ID	Description	Age of Adolescent (Years)
1	Mother & son	9
2	Mother & son	11
3	Mother & daughter	14
4	Adolescent female	14
5	Adolescent female	14
6	Mother & son	10
7	Mother	17
8	Mother	16
9	Father	10
10	Mother & son	15
11	Mother & son	16
12	Mother	13
13	Mother & daughter	12
14	Mother & son	10
15	Adolescent female	16
16	Adolescent female	13
17	Adolescent female	17
18	Adolescent male	10
19	Mother	11
20	Adolescent female	11
**Mean**		**12.95**

**Table 2 ijerph-19-06315-t002:** Initial coding example with data excerpt ID11 is shown in Figure 2.

Narrative	Initial Researcher Codes	Collaborative Codes with Co-Researchers	Theme Refinement
Adolescent ID11: “I use the phone to message my mum when I’m in my room or when she is out of the house. (Mum ID11: It’s a good thing because he can just message me). Adolescent ID11: “It makes me feel safer because I can use the phone to send my blood readings a to my mum. It is also good because it distracts me from my diabetes, and I can think about other things that are not to do with diabetes.”	Teen messaging mum—helps independence. Perceived as good by mum Feeling safe Messaging blood sugar data to mum when away Freedom from thinking about diabetes	Benefits of technology—communicating remotely about T1D for parental support Parental emotions: peace of mind Technology improves T1D perceived safety Teen resilience	Communication and immediate support (Benefits of technology) Emotional impact Resilience & positivity

## Data Availability

All data is stored on the secure University of Strathclyde OneDrive system. This data can be made available by contacting the lead author.

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
