# Peer review of "Letting the World See through Your Eyes: Using Photovoice to Explore the Role of Technology in Physical Activity for Adolescents Living with Type 1 Diabetes"

_ijerph, 2022, doi:10.3390/ijerph19106315_

Round 1

Reviewer 1 Report

General comments

The authors have clearly stated that the purpose of the study was to explore how technologies and physical activity are experienced by adolescents with type 1 diabetes. The paper is well-written and easy to follow. In my opinion, it adds considerable value to the current literature, since type 1 diabetes is rapidly increasing in this population while physical activity is considered a powerful tool for glucose control and insulin sensitivity. This study can enhance future attempts in similar research area in order to investigate more specific pathways between this metabolic disease and active lifestyle in youth. However, I have highlighted a few suggestions and concerns in my specific comments section (below) that need to be addressed before considering whether this work should be published or not. In general, the present paper is a novel work providing insightful information into the current literature.

Specific comments

ABSTRACT

  • Add a few sentences about type 1 diabetes and physical activity at the beginning.

  1. INTRODUCTION & DISCUSSION

- Nice work from the authors. I suggest providing a shorter introduction by                                                                                                                                  merging a few paragraphs to help readers follow the main messages provided in this section. Furthermore, add a few statements aiming to connect the topic with the current state of the health and fitness industry, and more specifically with the top relevant trends in this sector such as technology- and health-oriented trends appear to be very attractive in Europe and worldwide. Given that physical activity and exercise should be a critical piece of the diabetes management puzzle, please consider citing the following studies.

References:

  • Kercher VM, Kercher K, Bennion T, Levy P, Alexander C, Amaral PC, et al. 2022 Fitness Trends from Around the Globe. ACSMs Health Fit J 2022; 26(1): 21–37.
  • Batrakoulis A. European Fitness Trends for 2020. ACSMs Health Fit J 2019; 23(6): 28–35.

Author Response

ABSTRACT

  • Add a few sentences about type 1 diabetes and physical activity at the beginning.

Reply to point 1

Sentences added to abstract on lines 12 – 15  “Type 1 diabetes is a life-threatening autoimmune condition, which is highly prevalent in young children.  Physical activity is underutilized as part of treatment goals due to multifactorial challenges and lack of education in both the family setting and across society as a while.”

  1. INTRODUCTION & DISCUSSION

- Nice work from the authors. I suggest providing a shorter introduction by                                                                                                                                  merging a few paragraphs to help readers follow the main messages provided in this section. Furthermore, add a few statements aiming to connect the topic with the current state of the health and fitness industry, and more specifically with the top relevant trends in this sector such as technology- and health-oriented trends appear to be very attractive in Europe and worldwide. Given that physical activity and exercise should be a critical piece of the diabetes management puzzle, please consider citing the following studies.

Reply to point 2

Thank you for your kind review. 

Introduction has been shortened, original word count 1189, new word count 878.  Paragraphs merged and statements about current fitness trends have been added see lines 41- 49 for inclusion of Kercher citation also.

Reviewer 2 Report

Thank you for the opportunity to review this manuscript for IJERPH. It presents an interesting study, but several aspects could be improved, as described below. 

The introduction could provide a stronger foundation for the study/manuscript. In particular, I questioned the relevance/value of the paragraph about tangential technology for people with T1D (lines 109-123). 

The methods section could be tightened and the clarity improved in several locations. For example, there are parts that are repeated multiple times in different sections of the methods (e.g., how participants were recruited, describing participants in Tables 1 and 2, etc.). As well, it was unclear to me what the participants (co-researchers) were told to do exactly? Finally, the paper says there were 29 participants, but I believe details about only 20 are provided in Table 1 (and it would be helpful to sort this information somehow and summarize the gender and age data). 

The presentation of results was interesting and informative with the combination of photos and participant quotes. 

The discussion section provided a nice summary and interpretations of the four identified themes. However, the conclusions section was rather brief and I think the authors could expand upon the recommendations/implications further beyond the short bullets provided on lines 520-533 (including integrating relevant literature/references where useful). 

Again, thank you for the opportunity to review this manuscript. I wish the authors well with their future research in this area. 

Author Response

Reviewer 2

Thank you for the opportunity to review this manuscript for IJERPH. It presents an interesting study, but several aspects could be improved, as described below. 

The introduction could provide a stronger foundation for the study/manuscript. In particular, I questioned the relevance/value of the paragraph about tangential technology for people with T1D (lines 109-123). 

Response point 1

Thank you for your kind review.

Reduction of introduction and strengthened focus on lived experience and using individual experiences as the basis for intervention design and/or digital tools to support PA. Lines 109-123 “tangential technology” these lines have been deleted, as part of the restructuring and refocusing. Tangential software, ie the use of smartphones are prevalent in the movement of connected devices i.e. some closed-loop systems use a combination of Insulin pump (for dosing), continuous glucose monitor (connected to the pump via radiowave frequencies); and Smartphone for prompts, visible real-time data trends and sharing data.  Therefore, this element of technology is part of the Diabetes technology eco-system.  Lines 43-48 cover this concept in generic terms, as an introduction.   

The methods section could be tightened and the clarity improved in several locations. For example, there are parts that are repeated multiple times in different sections of the methods (e.g., how participants were recruited, describing participants in Tables 1 and 2, etc.). As well, it was unclear to me what the participants (co-researchers) were told to do exactly? Finally, the paper says there were 29 participants, but I believe details about only 20 are provided in Table 1 (and it would be helpful to sort this information somehow and summarize the gender and age data). 

Response to point 2

Clarity improved in methods section. Table 2 removed.  Table 1 updated to show participation description and gender/age descriptions.  Two tables summarised into one.

The presentation of results was interesting and informative with the combination of photos and participant quotes. 

The discussion section provided a nice summary and interpretations of the four identified themes. However, the conclusions section was rather brief and I think the authors could expand upon the recommendations/implications further beyond the short bullets provided on lines 520-533 (including integrating relevant literature/references where useful). 

Response to point 3

The conclusion has been extended and rewritten see lines 485-504

Again, thank you for the opportunity to review this manuscript. I wish the authors well with their future research in this area.

Reviewer 3 Report

The title “Letting the World See Through Your Eyes: Using Photovoice to 2 Explore the Role of Technology in Physical Activity for Adolescents Living with Type 1 Diabetes”. This manuscript engaged a qualitative method to explore how technologies and physical activity are experienced by adolescents with type 1 diabetes. It is very interesting to perform a study that used the qualitative method to deeply understand people’s feelings and thinking about the technology in physical activity intervention. However, there are some concerning should be answered:

  1. In the introduction part, should be short.
  2. In qualitative research, coding is very important. The coding of the analysis should be submitted as an attachment. Do you have used software to analyze the data, if you use it, please show the coding and memo?
  3. In qualitative research, the manuscript should tell a clear story about technology in physical activity. In the result part and discussion part, the author should show a clear thematic story.
  4. In the conclusion part, should be short and rewritten.

Author Response

Reviewer 3

The title “Letting the World See Through Your Eyes: Using Photovoice to 2 Explore the Role of Technology in Physical Activity for Adolescents Living with Type 1 Diabetes”. This manuscript engaged a qualitative method to explore how technologies and physical activity are experienced by adolescents with type 1 diabetes. It is very interesting to perform a study that used the qualitative method to deeply understand people’s feelings and thinking about the technology in physical activity intervention. However, there are some concerning should be answered:

  1. In the introduction part, should be short.
  2. In qualitative research, coding is very important. The coding of the analysis should be submitted as an attachment. Do you have used software to analyze the data, if you use it, please show the coding and memo?
  3. In qualitative research, the manuscript should tell a clear story about technology in physical activity. In the result part and discussion part, the author should show a clear thematic story.
  4. In the conclusion part, should be short and rewritten.

Response to point 1

Thank for your consideration.

The introduction has been reduced.

Response to point 2

I have attached examples of coding, which was done collaboratively with co-researchers.  There is also evidence of using NVivo software when naming themes. Explanation of using Photovoice through collaborative coding with participants is provided in lines 183-198.

Response to point 3

See re-focused introduction.  Results and discussion follow main theme’s which were interpreted in the data, thesea re written in response to the data.

Response to point 4

Conclusion has been changed.

Round 2

Reviewer 3 Report

Title” Letting the World See Through Your Eyes: Using Photovoice to Explore the Role of Technology in Physical Activity for Adolescents Living with Type 1 Diabetes”, the author has carefully revised the manuscript according to my suggestion.

 I have two concerns in the following:

1) Code process should show more details, not Nvivo coding figure but coding detail process. This is because it is very important for coding processing in photovoice analysis.

2) In the conclusion part, the conclusion should be shortly.

Author Response

Response to reviewer 3:

Additional analysis evidence has been submitted see lines 205 - 216.  See also amended Appendix, showing coding processes. 

Conclusion has been reduced in size.  Paragraph 1 was removed. Other reviewer asked for elaboration of bullet points in recommendations.